# Pyrophyllite: An Economic Mineral for Different Industrial Applications



**Maaz A. Ali** [1,*] , **Hussin A. M. Ahmed** [1,2] , **Haitham M. Ahmed** [1] and **Mohammed Hefni** [1]

1 Mining Engineering Department, King Abdulaziz University, Jeddah 21589, Saudi Arabia;
hussien135@gmail.com (H.A.M.A.); hmahmed@kau.edu.sa (H.M.A.); mhefni@kau.edu.sa (M.H.)
2 Central Metallurgical Research and Development Institute (CMRDI), Helwan 87, Cairo 11912, Egypt
* Correspondence: mnoureldaimali@stu.kau.edu.sa; Tel.: +966-557-370-521

**Abstract:** Pyrophyllite ($Al_2Si_4O_{10}(OH)_2$) is a phyllosilicate often associated with quartz, mica, kaolinite, epidote, and rutile minerals. In its pure state, pyrophyllite exhibits unique properties such as low thermal and electrical conductivity, high refractive behavior, low expansion coefficient, chemical inertness, and high resistance to corrosion by molten metals and gases. These properties make it desirable in different industries such as refractory; ceramic, fiberglass, and cosmetic industries; as filler in the paper, plastic, paint, and pesticide industries; as soil conditioner in the fertilizer industry; and as a dusting agent in the rubber and roofing industries. Pyrophyllite can also serve as an economical alternative in many industrial applications to different minerals as kaolinite, talc, and feldspar. To increase its market value, pyrophyllite must have high alumina ($Al_2O_3$) content, remain free of any impurities, and possess as much whiteness as possible. This paper presented a review of pyrophyllite's industrial applications, its important exploitable properties, and the specifications required for its use in industry. It also presents the most effective and economical techniques for enriching low-grade pyrophyllite ores to make them suitable for various industrial applications.

**Keywords:** pyrophyllite; properties; applications; specifications; enrichment; low-grade





## 1. Introduction

Pyrophyllite is a hydrous aluminum silicate with the chemical formula $Al_2Si_4O_{10}(OH)_2$ and is commonly associated with other minerals such as quartz, mica, kaolinite, epidote, and rutile [1–3]. The pure pyrophyllite is composed of 28.3% $Al_2O_3$, 66.7% $SiO_2$, and 5% $H_2O$ on weight bases [4]. Pyrophyllite, when pure, is desirable for many applications due to its unique properties [5]. For example, pyrophyllite provides low thermal and electrical conductivity; high refractive behavior; a low expansion coefficient; high corrosion resistance; low bulk density; and low hot-load deformation [6]. Therefore, this mineral is widely used in the refractory, ceramic, fiberglass, pesticide, fertilizer, paper, paint, plastic, rubber, cement, building-material, and pharmaceutical industries [7–9]. Furthermore, since pyrophyllite has a lower coefficient of expansion and thermal conductivity compared to clay, pyrophyllite is suitable for refractory applications [10,11]. Pyrophyllite, as a clay mineral, can be used as a substitute for kaolinite mineral group in many industrial applications, such as in the ceramic, and pottery, and filler industries, because kaolinite mineral group are rapidly depleting and expensive [12,13]. Furthermore, pyrophyllite can replace talc in many applications, especially as a filler pharmaceutical and medical applications, since pyrophyllite is safe and free from associated toxic minerals such as asbestos [14].

Determining the market price of pyrophyllite depends on the mineral's $Al_2O_3$ content and the presence of impurities (Table 1) [15]. Since pyrophyllite is formed via hydrothermal alterations, it contains impurities such as iron, titanium, and alkalis [16,17]. The quality of pyrophyllite products is greatly affected by these impurities [18]. In addition, the presence of iron (Fe) and titanium (Ti) in pyrophyllite causes discoloration of the final product [19]. Accordingly, when pyrophyllite is used in the ceramic industry, this mineral

stains the product's surface due to the presence of these impurities [12,20]. Moreover, small amounts of iron lead to a decrease in the melting point of the refractory materials, affect the transparency of glass products, and reduce transmission in optical fibers [1,3,8]. Therefore, the most desirable percentage of iron in the pyrophyllite ore for use in industrial applications was determined in a previous study [8]. For example, in the refractory industry, the ore should contain less than 1% iron; in pottery and tile manufacturing, the ore should contain less than 0.5% iron; and in the paper industry, the ore should contain less than 1% iron [21]. It should be noted that alkalis are also deleterious because alkalis in the form of carbonates combine with silica ($SiO_2$) under firing temperatures and form silicates that are soluble in water [3]. The presence of these silicates in products is not desirable, especially in the ceramic industry. Furthermore, alkali ions are electrically conductive [22,23].

**Table 1.** Pyrophyllite different grades [3,24–26].

| Grade | Main Specifications [3,24] | Price, US$/t [24–26] |
|---|---|---|
| Filler grade | 300 mesh, milled, 21–27% $Al_2O_3$ | 150–480 |
| Ceramic grade | 15–19% $Al_2O_3$ | 27–44 |
| Fiberglass grade | 18%-21% $Al_2O_3$ | 59–65 |
| Refractory grade | 18%-21% $Al_2O_3$ | 59–65 |

In 2017, a survey of the resource market showed significant growth in pyrophyllite production. High-grade pyrophyllite is produced in Korea, Japan, and China. Moreover, Japan and South Korea contribute more than 5% of the pyrophyllite produced globally [16,27]. Table 2 shows the world production of pyrophyllite (By Principal Countries) from 2015 to 2019. The increasing demand for high-grade pyrophyllite ores with low impurities for industrial applications led to a scarcity in the reserves of such ore [5,6]. Despite the presence of large quantities of low-grade pyrophyllite ores, the impurities in these ores limit their use. Thus, there is an imperative need for techniques to improve the purity of these low-grade ores [28].

**Table 2.** World pyrophyllite production in 2015–2019 in thousand metric tonnes (By principal countries) [26].

| Country | 2015 | 2016 | 2017 | 2018 | 2019 |
|---|---|---|---|---|---|
| Korea, Rep. of | 596.86 | 590.00 | 431.458 | 346.76 | 327.62 |
| Japan | 160.00 | 160.00 | 160.00 | 160.00 | 160.00 |
| India | 167.00 | 170.00 | 170.00 | 170.00 | 170.00 |
| Turkey | 50.00 | 50.00 | 50.00 | 50.00 | 50.00 |
| Thailand | 45.50 | 96.80 | 54.00 | 50.92 | 6.50 |
| Peru | 26.21 | 17.87 | 22.76 | 26.67 | 25.03 |
| South Africa | 17.35 | 19.11 | 55.04 | 98.24 | 134.45 |
| Saudi Arabia | 40.00 | 42.00 | 44.00 | 46.00 | 48.00 |

Several techniques have been tested to remove impurities from industrial minerals, such as removing iron from dolomite [29], talc [30], and kaolin [31,32]. Moreover, several studies have used electrolysis, flotation, and electrification to remove sulfur, chlorine, iron, and titanium from non-metallic minerals [20,28,33]. For pyrophyllite, gravity separation techniques, flotation, magnetic separation, and leaching by ammonia and oxalic acid have been explored [1,8,18,28,34,35]. Furthermore, microwaves combined with sequential magnetic separation treatment have been used and considered as a promising method for upgrading low-grade pyrophyllite ore [19,36]. Under this background, the present study reviews the industrial applications of pyrophyllite and its required specifications in addition to reviewing the most effective and economical methods for enriching low-grade pyrophyllite ores.

## 2. Unique Properties of Pyrophyllite

The unique chemical and physical properties of pyrophyllite make it suitable for use in many industrial applications and as a substitute for several minerals such as talc and kaolinite [37]. These properties are addressed in the following sections.

### 2.1. Chemical Properties

Pure pyrophyllite contains 66.7% $SiO_2$, 28.30% $Al_2O_3$, and 5.00% $H_2O^+$. However, high-purity pyrophyllite is rarely found in nature [38]. Table 3 shows the chemical composition of pure pyrophyllite in select deposits in South Korea and Russia [3]. One of the most important chemical properties that gives pyrophyllite a unique advantage in applied industries is that pyrophyllite is chemically inert and electrically neutral, making it highly resistant to the strongest acids and alkalis [39]. Pyrophyllite crystals can be either platy or fibrous (asbestine). Depending on their structures, these crystals break down into plates/flakes or fibers when ground. A sheet-like structure exists in each flake or folium, consisting of two silicate layers sandwiched by gibbsite [$Al(OH)_3$] layers. Pyrophyllite with a platy structure is desirable in industry, especially when used as a filler in the paint industry. This platy configuration, which is caused by the silicate structure of pyrophyllite sheets, increases resistance to film cracking, helps film dry, and promotes good dispersion [40].

**Table 3.** Pure pyrophyllite chemical composition in select deposits in South Korea and Russia [3,41,42].

| Chemical Composition % | South Korea | | | Russia | | |
|---|---|---|---|---|---|---|
| | Heanam | Gussi | Nohawado | Chistugor | Polar Urol | Kul-Yurt-Tau |
| $SiO_2$ | 66.0 | 66.2 | 67.6 | 67.0 | 66.9 | 66.5 |
| $Al_2O_3$ | 28.6 | 28.7 | 28.9 | 28.2 | 27.7 | 28.6 |
| FeO | 0.2 | 0.1 | 0.3 | NA | 0.3 | NA |
| $K_2O$ | 0.3 | NA * | 0.1 | 0.1 | NA | 0.1 |
| $Na_2O$ | 0.2 | 0.2 | 0.2 | NA | NA | 0.2 |

* NA: Not Available.

### 2.2. Physical Properties

Pyrophyllite has a hardness of 1–2 on the Moh's scale of hardness. Therefore, pyrophyllite is very soft and has a soapy and smooth surface, making it suitable for many industrial applications, such as in the pharmaceutical industry [7,43]. The high softness of pyrophyllite gives the mineral a high specific gravity of up to 2.9 [21]. Depending on the content of impurities such as iron oxide, the color of pyrophyllite varies from white to brownish green [43,44]. Moreover, it has a pearly luster in its foliated variety, with a refractive index that ranges from 1.55 to 1.60. Pyrophyllite also features good mechanical strength, thereby producing durable structures. Furthermore, pyrophyllite is a poor conductor of electricity and has high dielectric strength. One of the most important properties of pyrophyllite is its natural hydrophobicity [45]. This property is used to produce many roofing products such as sealing and waterproof materials and asphalt felts. In addition, ground pyrophyllite has a high oil absorption ability, especially the fibrous type of pyrophyllite. This property enables the use of pyrophyllite in the paint industry as an extender pigment [46].

### 2.3. Thermal Properties

Thermal properties represent the most important characteristic of pyrophyllite and combine the advantages of the thermal properties of both water and minerals. Like water (and unlike metal), pyrophyllite retains heat for a long time, radiating the heat slowly over an extended period, and, like metal, pyrophyllite heats up quickly. This behavior occurs because pyrophyllite combines poor thermal conductivity and high specific heat. Poor thermal conductivity means that the heat absorbed by the pyrophyllite does not flow quickly, while high specific heat means that if a unit rises in temperature, a mass of

pyrophyllite can absorb a large amount of heat. These properties enhance the durability of refractory products that use pyrophyllite [12,47,48].

### 2.4. Pyrophyllite Thermal Phase Transformation

Pyrophyllite, when heated at different temperatures, passes through several thermal transformation phases (Table 4). One of the most critical transformations is the conversion of pyrophyllite into mullite at a temperature above 1200 °C. Mullite is an aluminum silicate with the chemical formula $Al_{4+2x}Si_{2-2x}O_{10-x}$ and features distinctive technical properties. These properties include a low coefficient of expansion, low thermal conductivity, and resistance to corrosion from molten metals. Thus, mullite is used as a refractory material and as a substitute for feldspar and silica in ceramic applications, such as in floor and wall tiles [23,49].

**Table 4.** Pyrophyllite thermal phase transformation [50].

| Thermal Transformation Phase | Heating Temperature |
|---|---|
| Removal of water whether surface, in pores, and/or adsorbed | <450 °C |
| Dehydroxylation | 780 °C < T < 1100 °C |
| Formation of amorphous $SiO_2$ | 950 °C < T < 1100 °C |
| Formation of mullite and crystallization of cristobalite from amorphous $SiO_2$ | T > 1200 °C |

### 3. Pyrophyllite's Industrial Applications

Pyrophyllite is characterized by various chemical, physical, and thermal properties that make it suitable for several industrial applications. It has wide applications as a substitute for feldspar and silica due to its beneficial technical properties. The uses of pyrophyllite in industry include its application as a refractory material in the refractory industry and as a raw material in the ceramic, fiberglass, and cosmetic industries [51,52]. Moreover, pyrophyllite is used as filler in the paper, plastic, paint, and pharmaceutical industries [53]; as a soil conditioner in fertilizer applications; and as a dusting agent in the rubber and asphalt industries [54,55]. Figure 1 shows pyrophyllite consumption in different industries in the US [51]. This section discusses the applications, specifications, and grades of pyrophyllite ore required for different industries.

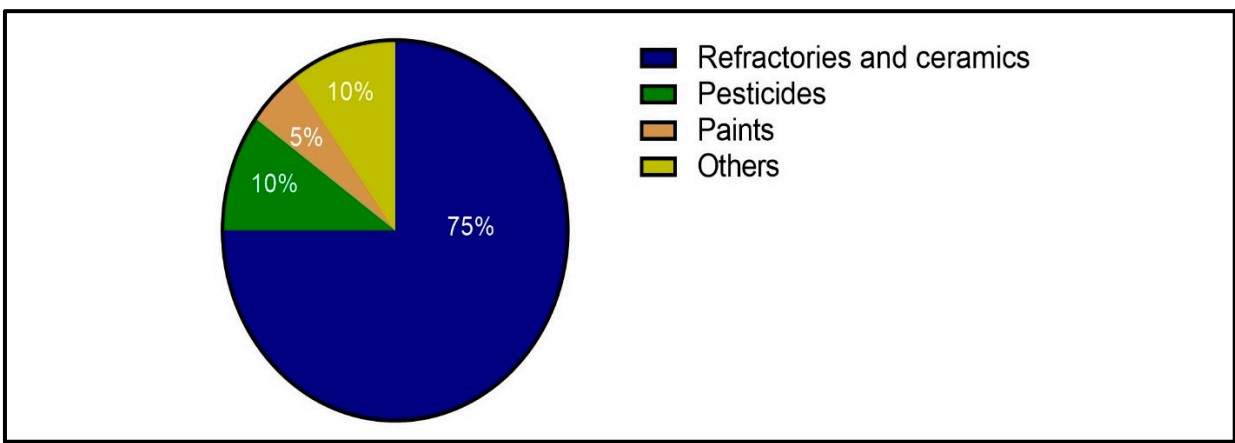

**Figure 1.** Pyrophyllite consumption by different industries in the US [51].

### 3.1. Pyrophyllite as a Refractory Material

Refractory Industry

Refractory materials are resistant to high temperatures and have a melting temperature of no less than 1580 °C, which is provided by pyrophyllite. Since pyrophyllite can be converted into mullite, pyrophyllite is suitable for producing relatively low-cost refractories, as mullite can withstand high temperatures up to 1810 °C [56]. Furthermore, pyrophyllite is highly resistant to chemical attacks by many silicates and gas oxides. In addition to its resistance to decrepitation, abrasion, and thermal shocks, pyrophyllite has high electrical resistance, compressive strength, tensile strength, and impact strength [57]. Therefore, pyrophyllite-based refractories are used in iron and steel furnaces for lining purposes. There are many ways to make pyrophyllite-based refractories. For example, (1) pyrophyllite can be crushed, bonded to sodium silicate, formed into bricks, and fired and (2) small amounts of pyrophyllite can be combined with fireclay, zirconia, or bauxite to make high-alumina refractories. Refractory products made from pyrophyllite include tile refractories, cement-fired bricks, fired brick-roofing tiles, and special refractories [45,58]. The specifications, grades, and properties of pyrophyllite required in the refractory industry are shown in Table 5.

### 3.2. Pyrophyllite as a Raw Material

3.2.1. Ceramic Industry

Clay minerals are the most important components in ceramic products. Moreover, several ceramic products can be prepared using different types of clay with different proportions in the mixture and different firing temperatures [59]. Since pyrophyllite becomes mullite under firing temperatures, which affords beneficial technical properties, pyrophyllite is used as a component in certain ceramic products. Ceramic products containing pyrophyllite include tiles, sanitary ware, white ware, and electrical components (e.g., insulators, vacuum gaskets, resistors, and transducers) [12,48]. The use of pyrophyllite in the ceramic industry improves mechanical properties, translucence, chemical resistance, resistance to thermal shocks; this mineral also provides high dielectric strength and promotes the crack-free glazing of the finished products. Further, pyrophyllite complements the silica in raw-material mixtures. Moreover, pyrophyllite's ability to heat quickly and convert into mullite at a lower temperature (1100 °C) enables faster firing cycles than similar materials [21].

According to the industrial specifications of ceramics, the percentages of impurities such as iron oxide, titanium oxide, and alkalis should remain within the permissible percentages because of their deleterious effects on products (Table 5) [51,60]. Iron and titanium also have a coloring effect on the final product. In addition, iron oxide with titanium forms a low-melting iron-titanate glass, which causes increased porosity in the final product through the formation of blisters [3,21,61]. Alkalis are also deleterious because alkalis in the form of carbonates combine with silica under firing temperatures to form water-soluble silicates. Additionally, alkali ions are electrically conductive [62].

3.2.2. Fiber Glass Industry

Pyrophyllite is used as an alternative to feldspar in the glass industry as a source of aluminum. Pure feldspar contains 18–19% alumina. Although pyrophyllite is used as a source of aluminum in the glass industry, it contains high alkali content, which causes problems in glass products, especially if the glass product is to be used in electrical and electronic devices [37]. Naturally occurring pyrophyllite contains about 19% alumina and is comparable to the content in feldspar. Moreover, pyrophyllite has low alkali content compared to feldspar, supporting its use in the glass industry. Pyrophyllite is used to prepare fiberglass batches and allows sand to be removed from the batch, thus improving the batch-to-melt conversion efficiency and reducing conversion energy [63]. Further, pyrophyllite provides excellent mechanical properties and low melting temperatures. The

pyrophyllite used in fiberglass production should be high in alumina content; low in iron oxide, titanium, and alkalis; and high in chemical stability, as shown in Table 5 [64,65].

### 3.2.3. Cosmetic Industry

Historically, talc has been used in the medical and pharmaceutical fields, especially in cosmetics, due to talc's desirable properties, such as chemical inertness and the lack of an environment conducive to the growth of bacteria [66]. However, concerns have been raised about talc being a carcinogen. These concerns stem from the fact that talc is naturally found in co-deposits with asbestos [67,68]. Pyrophyllite, with its unique properties suitable for the cosmetic industry, represents a safe alternative. Pyrophyllite has a natural white color, chemical inertness, smoothness, softness, and hydrophobic properties. The hydrophobic nature of pyrophyllite prevents the absorption of sweat and causes sticking. Accordingly, high-quality pyrophyllite is used as a base in various face and body powders [9,14,21]. The pyrophyllite used for this purpose should be very white, free from iron and other colored minerals, and lacking grit and calcite since these properties affect the smoothness of the final product (Table 5) [9].

### 3.3. Pyrophyllite as Filler

Mineral filler is ground rock added to a specific mixture to enhance performance and properties and reduce cost. Cost is reduced by replacing more expensive components with less expensive, more efficient mineral fillers that take up space in the product matrix. When choosing a filler material, factors that should be considered include the filler's chemical properties, cost, refractive index properties, particle size and shape, color, and hardness [69,70]. The most common mineral fillers are talc, pyrophyllite, mica, dolomite, calcium carbonate, wollastonite, and kaolin clay [70,71]. These fillers all provide cost savings but differ in their applications based on their properties. Due to its chemical and physical properties, pyrophyllite is very suitable as a filler and is thus used in the paper, plastic, paint, insecticide, and pharmaceutical industries.

### 3.3.1. Paper Industry

Papermakers seek to reduce costs and increase product quality through the use of fillers. The benefits of using fillers vary according to the product to be made. The most important characteristics of the final paper product are smoothness, brightness, opacity, print quality, dimensional stability, and low total cost. There are different types of fillers used in the manufacture of paper and can be either natural or synthetic. Natural fillers include ground limestone (ground chalk), kaolin, talc, and pyrophyllite, while synthetic fillers include precipitated calcium carbonate (PCC), precipitated aluminum silicate, titanium dioxide, and gypsum. Finely ground pyrophyllite is used as a filler in the paper industry because it is cheap and has excellent properties that improve the product quality, such as chemical inertness, softness, high reflectance, a strong particle shape, and hydrophobicity [3,72,73]. For this grade, the pyrophyllite should have high alumina content, low silica content, and a high degree of whiteness (Table 5).

### 3.3.2. Plastic Industry

Pyrophyllite is used in the plastic industry as a filler in various applications, such as in polyvinyl chloride (PVC), low-density polyethylene (LDPE), and high-density polyethylene (HDPE). In these applications, the various properties of pyrophyllite are exploited, such as its chemical inertness, high specific heat, high electrical resistance, platy structure, high oil absorption, greasiness, and good mechanical strength. These improved plastic materials are used in garden chairs and computer cases, which require high specific heat and electrical resistance, and under conditions that require high mechanical strength, such as in bumpers and automobile dashboards. In all these applications, pyrophyllite improves the smoothness and greasiness of the product. The platy structure of pyrophyllite gives the product a smooth finish. Further, pyrophyllite's high oil-absorption properties offer an

excellent mixture between pyrophyllite and oleo-resinous materials [9,21,74]. Moreover, pyrophyllite's chemical inertness prevents a chemical reaction from occurring between the materials. The industry in question will determine the size of the ultra-fine particles needed to achieve uniform dispersion within the matrix; the other specifications are shown in Table 5.

**Table 5.** Specifications of pyrophyllite ore for different applications.

| Industry (Role) | Specifications | | Ref. |
| --- | --- | --- | --- |
| | Chemical | Physical | |
| Refractory (as refractory material) | • $SiO_2$ 60% Max<br>• $Al_2O_3$ 18–21%<br>• $Fe_2O_3$ <1%<br>• $TiO_2$ 1% Max<br>• Alkalis 1% Max<br>• CaO <0.5%<br>• MgO <0.5%<br>• LOI 4% | • Required size<br>>150 mm (special refractory).<br>>15 mm (tile refractory).<br>>5 mm (cement brick).<br>• Density 2.8–2.9 g/cm³<br>• Pyrometric Cone Equivalent (PCE) 28–30% | [3,8,9,45] |
| Ceramic (as raw material) | • $SiO_2$ 64 ± 2%<br>• $Al_2O_3$ 15–19%<br>• $Fe_2O_3$ 1% Max<br>• $TiO_2$ 1% Max<br>• Alkalis 1% Max<br>• CaO <0.5%<br>• MgO <0.5%<br>• LOI 4.5 ± 1% | • Required size 44 Microns | [2,3,9,22] |
| Fiber glass (as raw material) | • $SiO_2$ <70%<br>• $Al_2O_3$ 18–21%<br>• $Fe_2O_3$ >0.5%<br>• $TiO_2$ >1%<br>• Alkalis >1%<br>• CaO <0.5%<br>• MgO <0.5%<br>• LOI 4% | • Required size −45 microns | [4,9,63,65] |
| Cosmetic (as raw material) | • $SiO_2$ <70%<br>• $Al_2O_3$ <21%<br>• $Fe_2O_3$ >0.5%<br>• $TiO_2$ >0.5%<br>• Alkalis >1%<br>• CaO <0.5%<br>• MgO <0.5%<br>• LOI 4% | • Required size Very fine (5 microns).<br>• Softness<br>• Smoothness | [3,14,21] |
| Paper (as filler) | • $SiO_2$ 55–70%<br>• $Al_2O_3$ 21–27%<br>• $Fe_2O_3$ >0.5%<br>• $TiO_2$ >0.5%<br>• Alkalis >1%<br>• CaO 1% Max<br>• MgO <0.5% | • Required size −45 microns<br>• Whiteness ≥ 85%<br>• Moisture 3% (min)<br>• Stable particle distribution | [3,9,21,72,75,76] |
| Plastic (as filler) | • $SiO_2$ <70%<br>• $Al_2O_3$ 21–27%<br>• $Fe_2O_3$ >0.5%<br>• $TiO_2$ >1%<br>• Alkalis 1% Max<br>• CaO >1%<br>• MgO <0.5% | • Required size −5 microns<br>• Density 2.8–2.9 g/cm³<br>• Mohs hardness, 1–2 | [3,9,51,72] |

**Table 5.** *Cont.*

| Industry (Role) | Specifications | | Ref. |
|---|---|---|---|
| | **Chemical** | **Physical** | |
| Paint (as filler) | • $SiO_2$ <65%<br>• $Al_2O_3$ <21%<br>• $Fe_2O_3$ >0.5%<br>• $TiO_2$ >1%<br>• Alkalis >0.5%<br>• CaO <0.5%<br>• MgO <0.5% | • Required size −53 microns<br>• Density 2.8–2.9 g/cm$^3$<br>• Mohs hardness, 1–2<br>• Brightness < 85%<br>• Oil absorption < 24% | [3,9,21,77,78] |
| Insecticide (as filler) | • $SiO_2$ <70%<br>• $Al_2O_3$ 19–21%<br>• $Fe_2O_3$ 1.5 Max<br>• $TiO_2$ >1%<br>• Alkalis >0.5%<br>• CaO <0.5%<br>• MgO <0.5%<br>• LOI −7% | • Required size −75 microns | [9,24,79–81] |
| Fertilizer (As soil conditioner) | • $SiO_2$ <70%<br>• $Al_2O_3$ 19–21% | • Required size 5 to 0.1 mm<br>• Mohs hardness 1.5 | [3,10,54,82,83] |
| Rubber (as dusting agent) | • $SiO_2$ >70%<br>• $Al_2O_3$ <19% | • Required size −75 microns (Max)<br>• Smooth, greasy feel | [7,9,51,71,84] |
| Roofing (as dusting agent) | • $SiO_2$ >70%<br>• $Al_2O_3$ <19% | • Required size −45 microns | [7,9,51,85,86] |

### 3.3.3. Paint Industry

High-quality finely ground pyrophyllite is used in paints as a pigment extender and suspending agent. Using pyrophyllite as an extender enhances the nature of a product by increasing the volume of paint when mixed with the product and increases resistance to film cracking. Moreover, pyrophyllite helps film dry and promotes good dispersion. The main properties of pyrophyllite exploited in the paint industry as an extender are the mineral's high refractive index, platy structure, softness, smooth surface, white color, chemical inertness, mechanical strength, and pearly luster. Another application of pyrophyllite in the paint industry is as a suspending agent in powder form. The fibrous crystalline type of pyrophyllite is used as a substitute for other, more expensive materials such as China clay. The fine fibers of pyrophyllite trap the primary pigment particles and keep them suspended long enough to facilitate brushing. In this way, the use of pyrophyllite ensures the pearly appearance, luster, and smooth brightness of the painted surface. Recently, finely ground paint-grade pyrophyllite has been used in wallboard joint cement and mastics to increase crack resistance and control rheology [3,51,53,87]. All these applications must consider the effects of iron oxide content, moisture, and volatile substances on the quality of the coating, along with the other specifications shown in Table 5.

### 3.3.4. Insecticide Industry

Pyrophyllite is used as a carrier in insecticides because it is cheap, fluffy, nonhygroscopic, and neutral in pH, making pyrophyllite compatible with both acid and alkaline ingredients. Other properties of pyrophyllite also make it desirable in this industry, such as the mineral's low water content, high specific gravity, and inertness. Further, pyrophyllite can be easily blown through a nozzle and quickly sticks to the leaves and stems of plants. When passing through the blower of a machine, an electrostatic charge is picked up, which attracts pyrophyllite to the undersides of the leaves and exposed upper plant

surfaces [9,24,54,79–81]. The most important industrial specifications for pyrophyllite to be used in insecticides are shown in Table 5.

### 3.4. As a Soil Conditioner

Recently, there has been increasing global interest in using aluminosilicate minerals to improve and maintain soil productivity. It is known that clay minerals can increase the ability of the soil to retain nutrients. The unique properties of pyrophyllite, such as its adsorption, ion exchange, and talc-like structure, have led to pyrophyllite's use in agriculture as a soil conditioner [54]. Pyrophyllite is used as a fertilizer carrier in agriculture, where it improves the ability of the soil to hold nutrients and reduces leaching. Furthermore, pyrophyllite, as an aluminosilicate mineral, is used to retain heavy metals in the soil due to the mineral's high surface area, high cation-exchange capacity, and large pore volume, allowing heavy metals to enter and be retained in its inner layers. Several studies have proven the ability of pyrophyllite to reduce the mobility of heavy metals in the soil, including Cu, Mn, Zn, Ni, Pb, Cd, and Cr. Pyrophyllite also contains certain amounts of potassium, calcium, magnesium, and iron, contributing significantly to plant growth and development. This grade of clay should contain a nearly neutral pH when mixed with other chemicals to provide stability over time (Table 5) [88].

### 3.5. As a Dusting Agent
#### 3.5.1. Rubber Industry

Low-grade pyrophyllite is used as a dusting agent in the rubber industry to reduce costs, lubricate molds, and prevent surfaces from adhering together during production. Several properties of pyrophyllite are exploited in these applications, such as its platy structure, chemical inertness, smoothness, and greasy feel, as these combined properties affect the quality and smoothing of rubber tires [9,21,55,89]. One of the most important specifications that must be considered is the fineness of the grains, and grit is highly undesirable (Table 5).

#### 3.5.2. Roofing Industry

Low-grade pyrophyllite is also used in the roofing industry. Pyrophyllite is added to asphalt roofing materials (i.e., in shingles and roll roofing) to prevent adhesion during manufacturing and storage and improve the materials' resistance to weathering. The size, fineness, chemical inertness, and high absorbency of pyrophyllite are among the most crucial criteria to be considered for this application (Table 5) [71].

### 4. Enrichment of Low-Grade Pyrophyllite Ore

Since high-grade pyrophyllite ores are rare globally, to be suitable for industry, low-grade pyrophyllite ores must be enriched to remove impurities such as Fe and Ti and increase the alumina content [37–39,52]. Techniques for upgrading low-grade pyrophyllite ore vary depending on the characterization of the outcomes; these methods include physical separation techniques, chemical separation techniques, and separation techniques that combine the two [8,19]. Physical separation techniques include dry/wet magnetic separation, attrition-scrubbing, and flotation. Chemical separation techniques include calcination/roasting and leaching using oxalic acid and an ammonia solution. Separation techniques that combine chemical and physical separation techniques include sequential calcination or roasting followed by magnetic separation [90–94]. These techniques are discussed based on researchers' contributions to the processing of low-grade pyrophyllite ore.

### 4.1. Magnetic Separation

Fe is a component in pyrophyllite that causes a decrease in the mineral's grade and affects the quality of the final product in terms of coloring and mechanical properties [95,96]. A magnetic separation process is one of the most important methods for removing Fe from low-grade pyrophyllite ore [27]. This technique can be performed either dry or wet. The

most important parameters considered when applying the magnetic separation method for low-grade pyrophyllite ore are the feed rate, feed% of solids, particle size, and magnetic intensity [97]. Several studies have investigated the use of magnetic separation to remove ferro and paramagnetic iron-bearing minerals from pyrophyllite. One study proved the efficiency of using magnetic separation to remove Fe from low-grade pyrophyllite. The studied samples contained pyrite and hematite phases, increased silicate content, and low alumina content. When high-intensity magnetic separation was applied, 97.6~98.8% Fe was removed from the treated samples [27]. In another study, dry magnetic separation with an intensity of 4000 Gauss was applied to low-grade pyrophyllite ore samples containing pyrite, hematite, and rutile phases (Table 6). The Fe removal rates obtained were 96% and 93% [36].

**Table 6.** Methods of beneficiation for low-grade pyrophyllite ores, description, and results obtained from the literature review.

| Ore Country | ROM Chemical Composition | Main Gangue Minerals | Treatment Methods | Major Concentrate Features | Ref. |
|---|---|---|---|---|---|
| South Korea | • $SiO_2$ 78.02% <br> • $Al_2O_3$ 10.12% <br> • $Fe_2O_3$ 2.99% <br> • $TiO_2$ 2.33% <br> • Alkalis 0.54 | • Hematite and pyrite phases | • Magnetic separation (Dry high intensity magnetic separation) | • The Fe removal rates obtained were 96% and 93%. | [98] |
| India | • $SiO_2$ 63.62% <br> • $Al_2O_3$ 21.36% <br> • $Fe_2O_3$ 0.26% <br> • $TiO_2$ 0.12% <br> • Alkalis 5.57% | • Quartz and feldspars | • Flotation (Direct) | • The $Al_2O_3$ content increased between 26.2 and 29.5%. <br> • The $SiO_2$ content decreased to 55.71% <br> • The product's brightness was enhanced. | [10] |
| Turkey | • $SiO_2$ 73.41% <br> • $Al_2O_3$ 20.64% <br> • $Fe_2O_3$ 0.06% <br> • $TiO_2$ 0.42% <br> • Alkalis 0.92% | • Quartz and kaolinite | • Collectorless <br> • Froth flotation (Direct) | • The $Al_2O_3$ content increased between 25 and 27%. <br> • The $SiO_2$ content decreased to 65.56%. | [94] |
| Turkey | • $SiO_2$ 73.41% <br> • $Al_2O_3$ 20.64% <br> • $Fe_2O_3$ 0.06% <br> • $TiO_2$ 0.42% <br> • Alkalis 0.92% | • Quartz and kaolinite | • Attrition-scrubbing | • A high $Al_2O_3$ grade of 29.33% was obtained alongside a decrease in $SiO_2$ content to 61% | [94] |
| India | • $SiO_2$ 68.13% <br> • $Al_2O_3$ 21.6% <br> • $Fe_2O_3$ 2.5% <br> • Alkalis 1.5% | • Quartz, muscovite, orthoclase, and goethite | • Leaching by oxalic acid | • The Fe removal rate obtained was 99.3%. | [8] |
| South Korea | • $SiO_2$ 71.64% <br> • $Al_2O_3$ 18.56% <br> • $Fe_2O_3$ 3.57% <br> • $TiO_2$ 0.57% <br> • Alkalis 0.48% | • Quartz and dickite. <br> • Euhedral cubic pyrites were observed | • Ammonia leaching solution | • A high Fe removal rate was obtained. | [35] |
| South Korea | • $SiO_2$ 72.37% <br> • $Al_2O_3$ 17.93% <br> • $Fe_2O_3$ 2.77% <br> • $TiO_2$ 1.39% <br> • Alkalis 1% | • Impurities appear in the form of pyrite, hematite, and rutile. <br> • Kaolinite, quartz, and dickite. | • Sequential microwave roasting and magnetic separation | • Iron and Ti were removed from the pyrophyllite ore with 86.3% and 68.3% efficiency, respectively. | [19] |

### 4.2. Flotation

The main objective of enrichment by flotation is to recover pyrophyllite from the associated minerals, thus increasing the $Al_2O_3$ content and decreasing the $SiO_2$ content [98,99]. Several studies have found that pyrophyllite responds to cationic collectors because of its lower aluminum: silica ratio, and its crystal structure contains the greatest number of cleavable planes. The most common flotation-based cationic collector for pyrophyllite is dodecylamine due to its satisfactory ability to collect aluminosilicate minerals at specific pH ranges [10,100,101]. The flotation of pyrophyllite was studied using dodecylamine and found to recover 96% of pyrophyllite. In a previous study, flotation was applied to pyrophyllite. In this study, the main associated minerals were quartz and feldspars (Table 6), and the collector was dodecylamine [101]. The study concluded that the $Al_2O_3$ content increased from 26.2 to 29.5%, the $SiO_2$ content decreased from 63.62% to 55.71%, and the product's brightness was enhanced [10]. Flotation of the pyrophyllite was also carried out using 3-diaminoprpopane and N-dodecyl-1, with a recovery rate higher than 80%. There were also attempts to use an anionic collector for the flotation of pyrophyllite ore containing quartz as a gangue mineral. It was found that using sodium oleate (SO) as a collector can improve the flotation efficiency of pyrophyllite [38].

It should be highlighted that the hydrophobic surfaces of pyrophyllite serve as natural adsorption sites for non-polar organic molecules. As a result of its hydrophobicity, pyrophyllite can be easily separated from quartz and feldspar using collectorless flotation and a single type of frother. Based on this phenomenon, a study was conducted using a collectorless froth flotation to enrich pyrophyllite using Methyl Isobutyl Carbinol (MIBC) as a frother reagent. The associated minerals were quartz and kaolinite (Table 6). The study concluded that using collectorless flotation to upgrade the pyrophyllite with different doses of MIBC produced an increase in $Al_2O_3$ content between 25% and 27% and decreased the $SiO_2$ content from 73.41% to 65.56% [94]. The XRD analysis in Figure 2 shows that the major crystalline phase was pyrophyllite, which increased via flotation with MIBC compared to raw pyrophyllite in which quartz was a major phase.

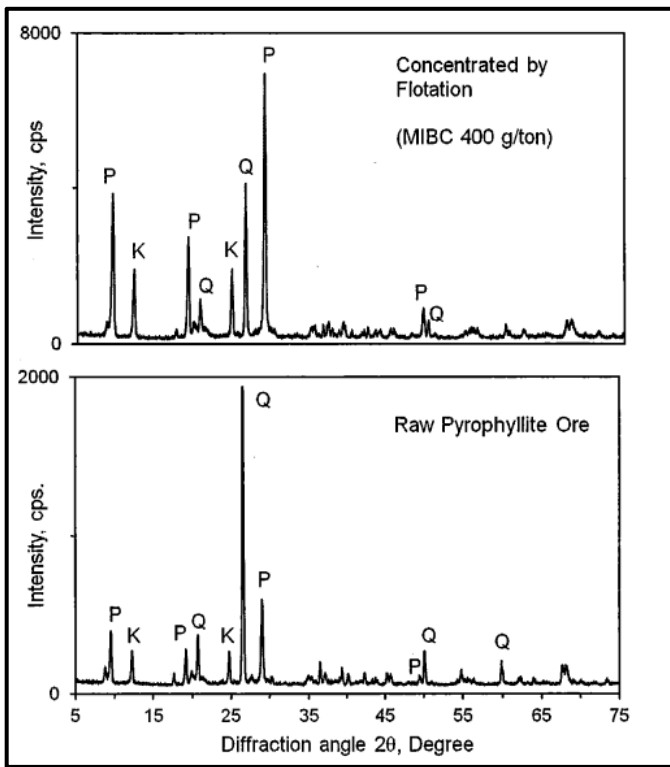

**Figure 2.** Comparison of the XRD patterns of raw pyrophyllite and the selected pyrophyllite concentrate obtained via collectorless froth flotation. P: pyrophyllite; Q: quartz; K: kaolinite [94].

In other cases, the flotation of pyrite from pyrophyllite was studied using N-dodecyl mercaptan as a collector along with a novel depressant (i.e., glucan) to achieve the selective flotation of pyrite and depress the pyrophyllite minerals. The studies demonstrated that a high recovery of Fe up to 98.51% could be obtained [98,102].

*4.3. Attrition-Scrubbing*

The Attrition-scrubbing technique is a simple beneficiation process in which the mineral particle is scrubbed under a high-slurry-flow speed, which also allows the particles to be affected by each other. This attrition leads to friction and collisions between the particles themselves. The attrition cell walls, impellers, and deflectors cause scrubbing, abrasion, and particle disintegration. This technique has wide applications for removing clay minerals from low-grade ores and has proven to be efficient in enriching several mineral ores, such as uranium and sand [103]. Moreover, this technique was shown to effectively concentrate low-grade pyrophyllite ore by significantly increasing the $Al_2O_3$ content and decreasing the $SiO_2$ content [104]. In another study, the attrition-scrubbing method was used to upgrade pyrophyllite ore that contains quartz and kaolinite as gangue minerals (Table 6). The study showed a significant improvement in the $Al_2O_3$ grade, where a very high $Al_2O_3$ grade of 29.33% was obtained for the finest size (−75 microns) compared to 21% for the $Al_2O_3$ grade in the feed, alongside a decrease in $SiO_2$ content to 61%, indicating high purity rates for pyrophyllite. Further, this study indicated that the attrition-scrubbing technique is effective for separating quartz and kaolinite from pyrophyllite [94]. Figure 3 presents an XRD analysis showing that the pyrophyllite was concentrated at a size of 75 microns under attrition-scrubbing. The major crystalline phase was pyrophyllite, which was increased via scrubbing compared to raw pyrophyllite, in which quartz was a major phase. Attrition-scrubbing clearly gives excellent results in separating clay minerals associated with pyrophyllite, which in turn improves the processing economy and energy consumption at the level of industry.

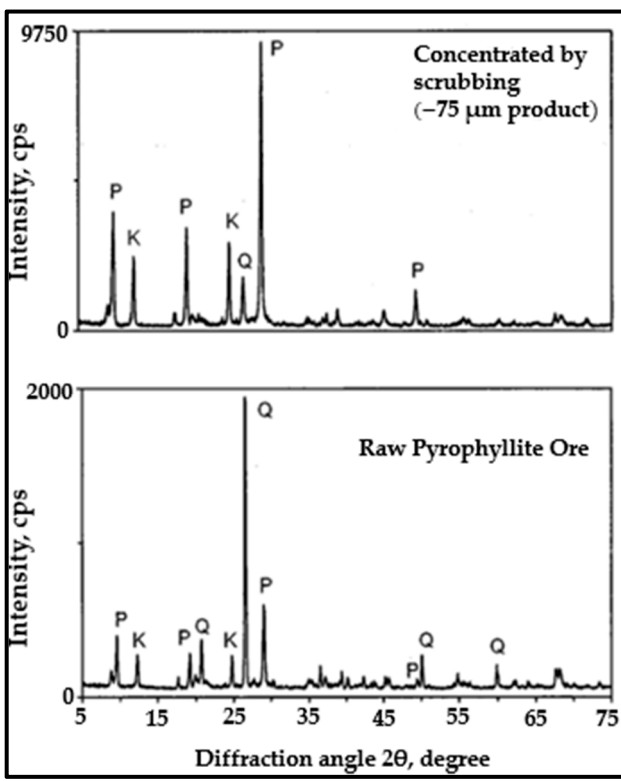

**Figure 3.** Comparison of the XRD patterns of raw pyrophyllite and selected pyrophyllite concentrates obtained by attrition-scrubbing. P: pyrophyllite; Q: quartz; K: kaolinite [94].

### 4.4. Leaching

In some cases, chemical treatment is used when physical treatment does not remove iron contaminants effectively from low-grade pyrophyllite ore. Leaching is the most important chemical treatment process for clay minerals and can use either organic acids (e.g., citric acid and oxalic acid) or inorganic acids (e.g., hydrochloric acid, sulfuric acid, and sodium hypochlorite). However, there are limitations when using inorganic acids with clay minerals due to environmental pollution and the contamination of products with Cl and $SO_4^{-2}$. Therefore, organic acids are more widely used. Oxalic acid is preferred for dissolving iron contaminants in clay minerals because of its high leachability under different conditions [105,106]. The effectiveness of oxalic acid in dissolving Fe from low-grade pyrophyllite ore to improve the quality of pyrophyllite was investigated in previous studies [8,107,108]. The critical parameters affecting the dissolution of Fe from pyrophyllite ore are the particle size, acid concentration, solid/liquid ratio, leaching time, temperature, and stirring speed. A study was also conducted to remove iron from pyrophyllite using oxalic acid leaching. The main gangue minerals were quartz, muscovite, and orthoclase (Table 6). Fe, mostly goethite, was present as inclusions or intergranular spaces within the silicates. When oxalic acid was used to dissolve iron using a concentration of 0.3 M, a temperature of 90 °C, a pulp density of 5%, a particle size below 100 microns, and a leaching time of 60 min, it was able to remove up to 99.3% of the iron in the pyrophyllite [8]. Oxalic acid is utilized with other chemical processes such as calcination, filtering, and drying to improve the quality of pyrophyllite by increasing its whiteness and brightness. These processes were applied to pyrophyllite micro powder with a particle size of 325–1200 meshes, and whiteness of greater than 87% was obtained [107,108].

There were previous attempts to use ammonia as a solvent to improve the quality of low-grade pyrophyllite ore via the dissolution of iron because ammonia is characterized by a low rinsing cost, low toxicity, and high efficiency in separating iron components [35]. In a previous study, the effectiveness of using an ammonia solution to remove iron from pyrophyllite was further investigated. The associated gangue minerals were quartz and dickite, and euhedral cubic pyrites were also observed (Table 6). The effect of variables such as the ammonium sulfate amount, particle size, addition of hydrogen peroxide, and sulfuric acid concentration were investigated. The study concluded that iron removal using an ammonia leaching solution could be effective [35].

### 4.5. Microwave Roasting and Magnetic Separation

Chemical separation methods are often applied alongside physical separation methods to obtain a higher-purity product [109]. Recently, the efficiency of using microwave heating and magnetic separation in removing impurities (e.g., Fe and Ti) from pyrophyllite was investigated and considered a promising and environmentally friendly method for enriching low-grade pyrophyllite ore [36]. Through previous studies, the use of microwave roasting and magnetic separation removed up to 96% of pyrophyllite impurities based on operating variables such as irradiation time and magnetic field intensity [36]. An increase in impurity removal efficiency was caused by phase changes of the impurities, which became magnetized during microwave roasting. Additionally, the impurity-removal efficiency can be improved by adjusting the operating conditions related to roasting and magnetic separation. Another study confirmed that sequential microwave roasting and magnetic separation could remove Fe and Ti with high efficiency. Kaolinite, quartz, and dickite were the main gangue minerals, and impurities in pyrophyllite occurred in the form of oxide and sulfide minerals. The study results indicated that Fe and Ti were removed from pyrophyllite with 86% and 68% efficiency, respectively, under 30 min of microwave irradiation and a magnetic field intensity of 2000 Gauss. Moreover, the study found that extending the microwave irradiation time and increasing the magnetic field intensity could improve impurity-removal efficiency, especially for paramagnetic Ti impurities [19]. This method can effectively upgrade low-grade pyrophyllite ore and clay minerals, which can then be exploited after removing impurities. However, to use this technique in industry,

more studies are needed to better optimize the mineral-phase changes to achieve effective separation and energy consumption.

## 5. Conclusions

Pyrophyllite is a hydrated silicate mineral with unique properties that make it easily processable and suitable as a substitute for several clay minerals, such as kaolinite, talc, and feldspar, in many industrial applications. The properties of pyrophyllite can be exploited to use the mineral as a refractory material in the refractory industry; as a raw material in the ceramic, fiberglass, and cosmetic industries; as a filler in the paper, plastic, paint, and pesticide industries; as a soil conditioner in the fertilizer industry; and as a dusting agent in the rubber and roofing industries. These industries require particular specifications of pyrophyllite to use the mineral, the most important of which is the grade of $Al_2O_3$ and the content of impurities. The grade of $Al_2O_3$ and the content of impurities determine the price of pyrophyllite, so obtaining high-alumina and low-impurity pyrophyllite is a goal for industrial applications. Since high-grade pyrophyllite is rare worldwide, low-grade pyrophyllite beneficiation is necessary to obtain a suitable product for industry. Enrichment methods for pyrophyllite aim to increase the alumina content and remove objectionable impurities (e.g., Fe and Ti). Techniques for improving low-grade pyrophyllite ore vary depending on the characterization outcomes and include physical separation techniques, chemical separation techniques, and separation techniques that combine the two. The most vital and efficient physical separation methods to remove impurities from low-grade pyrophyllite are magnetic separation, flotation, and attrition-scrubbing, mainly when the gangue minerals are quartz and clay minerals. Previous studies found that these physical separation techniques are more efficient and commercially viable for low-grade pyrophyllite ores. In terms of chemical methods, dissolution with oxalic acid works to remove iron efficiently and increase the degree of whiteness of the pyrophyllite when the process is carried out alongside other chemical processes, such as calcination. Finally, based on previous studies, it is clear that sequential microwave roasting with magnetic separation is a promising, environmentally friendly, and economical method for enriching low-grade pyrophyllite ore and other clay minerals.

**Author Contributions:** Conceptualization, M.A.A. and H.A.M.A.; formal analysis, M.A.A.; resources, H.M.A.; data curation, M.A.A., H.A.M.A., H.M.A. and M.H.; writing—original draft preparation, M.A.A.; writing—review and editing, H.A.M.A., H.M.A. and M.H.; visualization, H.A.M.A., H.M.A. and M.H.; supervision, H.A.M.A.; funding acquisition, H.A.M.A., H.M.A. and M.H. All authors have read and agreed to the published version of the manuscript.

**Funding:** This work was supported by the Deanship of Scientific Research (DSR), King Abdulaziz University, Jeddah, under grant No. D1434-135-002. The authors, therefore, gratefully acknowledge the DSR's technical and financial support.

**Institutional Review Board Statement:** Not applicable.

**Informed Consent Statement:** Not applicable.

**Data Availability Statement:** Not applicable.

**Acknowledgments:** This project was funded by the Deanship of Scientific Research (DSR), King Abdulaziz University, Jeddah, under Grant No. (No. D1434-135-002). The authors, therefore, gratefully acknowledge DSR technical and financial support.

**Conflicts of Interest:** The authors declare no conflict of interest.

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
