# Peer review of "Pyrophyllite: An Economic Mineral for Different Industrial Applications"

_applsci, doi:10.3390/app112311357_

Round 1
Reviewer 1 Report
Dear Authors,
The reviewed manuscript is a compilation of data from various scientific publications. The authors do not submit their data. I have mixed feelings about the manuscript under review, on the one hand, it is written correctly, but the information is presented at a high level of generality, which may be due to the lack of experience of researchers in the study of pyrophyllite, as evidenced by the lack of citation of own research results and own papers on this topic. For me, the publication is important only as a structured material useful for teaching activities. If there is a need to collect pyrophyllite data, I would prefer to use original papers. I leave the final decision accepting the publication of the article to the volume editor, in the evaluation system I choose the minor revision option. Below, I have highlighted a few inaccuracies that should be corrected. The text is understandable, although I am not a native speaker and I cannot judge the beauty of the English language used.
line 24: "kaolin" is not the name of the mineral
line 25, 26....: "%" of what? - modify throughout the manuscript, please
line 26: "as a phyllosilicate" - remove it, please
line 34, 36: "kaolinitic minerals" please change to "kaolinite mineral group"
line 130: use proper crystal-chemical formula of mullite //Al4+2xSi2-2xO10-x
(x ~ 0.4)//
line 135: "Pyrophyllite", please start explanation with capital letter
figure 2: This figure is not necessary for the article. It can be removed.
Table 5: unsightly, unclear, scattered distribution of data in the table. Please reformat the table.
lines 345-350: Only iron minerals and not Fe can be removed using magnetic methods. Pyrite, unless weathered, is a non-magnetic mineral and cannot be removed by magnetic separation techniques. Please correct and/or clarify this fragment of the manuscript.
line 412: L...../capital letter/
Table 6: What do the oxide contents in the table mean. It is the composition of the ore before or after enrichment. Two of these results would be worth giving. The layout of the table is unfriendly to the reader, please change it.
Figure 5. This figure is not necessary. It can be removed.
Author Response
First of all, we would thank Reviewer 1 for his constructive comments and opinions, which significantly improved the paper, and thank him for his time and effort with us.
Please see the attachment

Reviewer 2 Report
Do you have any publications or even unpublished research data on the topic you are reviewing?
Line 7, 14 and thereafter - kaolin is not a mineral species but the name of soft white clay composed mainly of the mineral kaolinite. So, avoid using kaolin as a mineral name.
Line 79 (Table 2) - What unit is used in this table? Thousand metric tons? Write in the table caption.
Line 83 - kaolin is not mineral!
Line 85 - 66.7% SiO2, if you have to be precise. Correct it.
Line 98-100 (Table 3) - No need to show unavailable TiO2 and CaO, just remove these rows from the table.
Line 148 (Fig. 1) - Data from 1996, no newer ones?
Line 422 - Correct to: previous studies
Line 477 - kaolin is not mineral!
References should be corrected strictly according to the Instructions for Authors! Too many errors and incomplete attributes in the Reference List...
Author Response
First of all, we would thank Reviewer 2 for his constructive feedback and opinions, which significantly improved the paper, and thanks go to him for his time and effort with us.
